# Glycemic Control Assessed by Intermittently Scanned Glucose Monitoring in Type 1 Diabetes during the COVID-19 Pandemic in Austria

**DOI:** 10.3390/s24144514

**Published:** 2024-07-12

**Authors:** Katharina Secco, Petra Martina Baumann, Tina Pöttler, Felix Aberer, Monika Cigler, Hesham Elsayed, Clemens Martin Harer, Raimund Weitgasser, Ingrid Schütz-Fuhrmann, Julia Katharina Mader

**Affiliations:** 1Division of Endocrinology and Diabetology, Department of Internal Medicine, Medical University of Graz, Auenbruggerplatz 15, 8036 Graz, Austria; katharina.secco@icloud.com (K.S.); petra.baumann@medunigraz.at (P.M.B.); tina.poettler@medunigraz.at (T.P.); felix.aberer@medunigraz.at (F.A.); monika.cigler@medunigraz.at (M.C.); hesham.elsayed@medunigraz.at (H.E.); clemens.harer@medunigraz.at (C.M.H.); 2Department of Internal Medicine and Diabetology, Private Clinic Wehrle-Diakonissen, 5026 Salzburg, Austria; raimund.weitgasser@diabeteszentrum-salzburg.at; 33rd Medical Division for Metabolic Diseases and Nephrology, Hospital Hietzing, 1130 Vienna, Austria; ingrid.schuetz-fuhrmann@gesundheitsverbund.at; 4Institute for Metabolic Diseases and Nephrology, Karl-Landsteiner Institute, 1130 Vienna, Austria

**Keywords:** COVID-19, glycemic control, isCGM, lockdown, pandemic, type 1 diabetes

## Abstract

Objective: The aim of this analysis was to assess glycemic control before and during the coronavirus disease (COVID-19) pandemic. Methods: Data from 64 (main analysis) and 80 (sensitivity analysis) people with type 1 diabetes (T1D) using intermittently scanned continuous glucose monitoring (isCGM) were investigated retrospectively. The baseline characteristics were collected from electronic medical records. The data were examined over three periods of three months each: from 16th of March 2019 until 16th of June 2019 (pre-pandemic), from 1st of December 2019 until 29th of February 2020 (pre-lockdown) and from 16th of March 2020 until 16th of June 2020 (lockdown 2020), representing the very beginning of the COVID-19 pandemic and the first Austrian-wide lockdown. Results: For the main analysis, 64 individuals with T1D (22 female, 42 male), who had a mean glycated hemoglobin (HbA1c) of 58.5 mmol/mol (51.0 to 69.3 mmol/mol) and a mean diabetes duration 13.5 years (5.5 to 22.0 years) were included in the analysis. The time in range (TIR_[70–180mg/dL]_) was the highest percentage of measures within all three studied phases, but the lockdown 2020 phase delivered the best data in all these cases. Concerning the time below range (TBR_[<70mg/dL]_) and the time above range (TAR_[>180mg/dL]_), the lockdown 2020 phase also had the best values. Regarding the sensitivity analysis, 80 individuals with T1D (26 female, 54 male), who had a mean HbA1c of 57.5 mmol/mol (51.0 to 69.3 mmol/mol) and a mean diabetes duration of 12.5 years (5.5 to 20.7 years), were included. The TIR_[70–180mg/dL]_ was also the highest percentage of measures within all three studied phases, with the lockdown 2020 phase also delivering the best data in all these cases. The TBR_[<70mg/dL]_ and the TAR_[>180mg/dL]_ underscored the data in the main analysis. Conclusion: Superior glycemic control, based on all parameters analyzed, was achieved during the first Austrian-wide lockdown compared to prior periods, which might be a result of reduced daily exertion or more time spent focusing on glycemic management.

## 1. Introduction

The increased blood sugar levels related to diabetes mellitus (DM) are a decisive factor in the development of micro- and macrovascular complications and diseases. Regarding microvascular disease nephropathies, neuropathies and retinopathies are the most common issues. Concerning macrovascular disease, organ tissue damage caused by peripheral vascular diseases, cerebrovascular disease and ischemic heart disease are the main causes [1,2,3,4,5]. For this reason, therapy for type 1 diabetes (T1D) aims to normalize the blood glucose (BG) levels, thereby avoiding acute and subsequent complications, achieving reductions in symptoms and, thus, restoring or maintaining quality of life [1,2,3,6,7,8,9,10]. The goal is to achieve glucose levels between 70 and 180 mg/dL for more than 70% of the time in range (TIR_[70–180mg/dL]_) [11,12]. The time above range (TAR_[>180mg/dL]_) should be <25%, and the time below range (TBR_[<70mg/dL]_) should be <5%. Specifically, <4% should be 54–<70 mg/dL, and <1% should be <54 mg/dL. Intermittently scanned continuous glucose monitoring (isCGM) offers comprehensive information on glucose variability and trends. Diabetes management can thereby be individualized by clinicians on the basis of continuous data of diabetes patterns [11,13,14,15]. Research in isCGM has grown rapidly in the last several years, but with the pandemic, things have changed. In Austria, as in many other countries, routine clinical appointments were cancelled during the coronavirus disease (COVID-19) pandemic. This was required, on the one hand, to shift resources to the acute care of patients infected with COVID-19 and on the other hand, to reduce potential COVID-19 infections in people with chronic conditions such as diabetes [16,17,18,19,20,21]. As the data suggested that people with inadequate glycemic control suffer from a more severe COVID-19 disease progression and are at a higher risk of death, it was essential to achieve good glycemic control, even when access to physicians for routine care was limited [1,6,7,8,10,22]. Generally, the consequences of the COVID-19 pandemic on people are currently under investigation worldwide. It is of highly importance to find out what these consequences are, especially in certain groups of people. In the current study, we wanted to have a closer look at the effects of COVID-19 on people living with T1D, as DM affects 1 out of every 11 people worldwide and was the ninth leading cause of death in 2019 [1,9,10,12,21,23,24,25]. As there still is a lack of research in this field, especially in Austria, we provide data obtained by isCGM, which combines the fields of chemistry and medicine. It should provide information in both fields—COVID-19 research and T1D research.

## 2. Materials and Methods

Data from people with T1D using isCGM and the Abbott Freestyle Libre system (Abbott Diabetes Care, Alameda, CA USA) [26] were investigated retrospectively within our study. Within this cohort study, the period from 16th of March in 2019 to 16th of June 2019 (pre-pandemic) was compared to the period from 1st of December 2019 to 29th of February 2020 (pre-lockdown) and from 16th or March in 2020 to 16th of June in 2020 (lockdown 2020). These periods represented the very beginning of the COVID-19 pandemic and the first Austria-wide lockdown. Because of the exploratory nature of this study, descriptive analyses were performed. The main analysis only included participants with isCGM data from all three periods, as well as data on baseline characteristics. For each period, the following isCGM-derived parameters were analyzed: isCGM activity, mean glucose (mg/dL), coefficient of glycemic variation (CV), glucose management indicator (GMI), mean amplitude of glycemic excursions (MAGE), time in different ranges (<54 mg/dL, 54–<70 mg/dL, 70–180 mg/dL, >180–250 mg/dL and >250 mg/dL, respectively), as well as total and mean daily scan frequency. Summary statistics of the parameters are presented for each period. Additionally, differences between the two prior periods and the lockdown period were calculated for each participant and then summarized.

The isCGM activity is based on a nominal number of 96 isCGM values per day (every 15 min = 24×4 measurements per day). Thus, for each individual *i* in each phase *p*, the total nominal number of measurements (*n_nom_*) is the number of available days (*n_days_*) multiplied by 96. The isCGM activity (*activity_isCGM_* [%]) for each individual *i* in phase *p* is then the percentage of the actual number of measurements (*n_act_*):(1)$${Activity}_{isCGM}^{ip}=\frac{n_{act}^{ip}}{n_{nom}^{ip}}*100$$
where
(2)$$n_{nom}^{ip}=n_{days}^{ip}*96$$

Mean glucose (*Mean_gluc_*) is the unweighted average of all glucose measurements (*G_*1*…j_*) for an individual *i* in phase *p*:(3)$${Mean}_{gluc}^{ip}=\frac{1}{n_{ipj}}\sum_{j=1}^{n_{ipj}}G_{ipj}$$

The coefficient of glycemic variation (*CV_gluc_*) for an individual *i* in phase *p* is the standard deviation (*SD_gluc_*) of glucose measurements (*G_*1*…j_*) divided by *Mean_gluc_*:(4)$${CV}_{gluc}^{ip}=\frac{{SD}_{gluc}^{ip}}{{Mean}_{gluc}^{ip}}$$
where
(5)$${SD}_{gluc}^{ip}=\sqrt{\;\frac{1}{\left(n_{ipj}-1\right)}\sum_{j=1}^{n_{ipj}}{\left(G_{ipj}-{Mean}_{gluc}^{ip}\right)}^{2}}$$

The glucose management indicator (*GMI*) for an individual *i* in phase *p* is calculated by the formula given by Bergenstal et al. [27].
(6)$${GMI}^{ip}=3.31+0.02392*\;{Mean}_{gluc}^{ip}$$

The mean amplitude of glycemic excursions (*MAGE*) of an individual *i* in phase *p* is the average of glucose values more than one SD away from the mean (amplitudes of glycemic excursions, *AGE*):(7)$${AGE}^{ip}=\left\{\begin{array}{l} G_{ipj},\;\;if\;\left|G_{ipj}-{Mean}_{gluc}^{ip}\right|>{SD}_{ip}^{gluc}\\ NA,\;\;otherwise \end{array}\right.$$
(8)$${MAGE}^{ip}=\frac{\sum_{k=1}^{n^{{AGE}^{ip}\;not\;NA}}{AGE}^{ip}}{n^{{AGE}^{ip}\;not\;NA}}$$

The time in range (*TIR[a−b]*) for an individual *i* in phase *p* is the percentage of values within a specific range a to b:(9)$${TIR}_{\left[a–b\right]}^{ip}=\left(\frac{1}{n_{ipj}}\sum_{j=1}^{n_{ipj}}Ι\left(a\leq G_{ipj}\leq b\right)\right)*100$$

The time above range (*TAR*) for an individual *i* in phase *p* is the percentage of values above a specific threshold a:(10)$${TBR}_{a}^{ip}=\left(\frac{1}{n_{ipj}}\sum_{j=1}^{n_{ipj}}Ι\left(G_{ipj}>a\right)\right)*100$$

The time below range (*TBR*) for an individual *i* in phase *p* is the percentage of values below a specific threshold b:(11)$${TBR}_{b}^{ip}=\left(\frac{1}{n_{ipj}}\sum_{j=1}^{n_{ipj}}Ι\left(G_{ipj}<b\right)\right)*100$$

The total scan frequency for an individual *i* in phase *p* is the number of actual glucose values ${(n}_{act}^{ip}),$ and the mean daily scan frequency is the average number of daily glucose values ($n_{act}^{ipd})$:(12)$${Mean\;daily\;scan\;frequency}_{ip}=\frac{1}{n_{days}^{ip}}\sum_{d=1}^{n_{days}^{ip}}n_{act}^{ipd}$$

Summaries of baseline characteristics are presented as Median (Q1–Q3) for numeric variables and N (%) for categorical variables. Summaries of isCGM-derived parameters and times in ranges are presented as median (Q1–Q3) and min-max.

A sensitivity analysis was conducted adding people with inconsistent data. Here, all individuals for whom isCGM data from the lockdown 2020 phase plus at least one other period, as well as the baseline characteristics that were available, were included.

The registry was submitted and approved by the Ethics Committee of the Medical University of Graz (ethics number 29-522 ex 16/17). All analyses were performed within the scope of the registry study.

## 3. Results

### 3.1. Main Analysis

A total of 64 people with T1D (Appendix A) were included in the study. The diabetes duration was 13.5 years (5.5; 22.0 years) and the baseline HbA1c was at 58.5 mmol/mol (51.0; 69.3 mmol/mol). They were used to the isCGM system, with an average use of 5.7 years (4.8; 6.6 years) prior to this study and had fewer visits during lockdown than during prior phases (Table 1).

The lockdown 2020 phase showed the best mean glucose data, with mean glucose values of 163.6 mg/dL (154.0–193.0 mg/dL). The data from 64 individuals were analyzed and, in general, the isCGM parameters were similar across all three phases. The isCGM values (n) were nearly the same in the pre-pandemic 2019 phase and in the lockdown 2020 phase and showed just a small difference during the pre-lockdown 2020 phase (Table 2).

To make a statement regarding the comparison of the respective phases, the differences between the periods (lockdown 2020 vs. pre-pandemic 2020 and lockdown 2020 vs. pre-lockdown 2020) were analyzed (Table 3). The lockdown 2020 vs. pre-lockdown 2020 comparison showed more differences than the lockdown 2020 vs. pre-pandemic 2019 comparison.

The TIR_[70–180mg/dL]_ was the highest percentage across all phases, but the lockdown 2020 phase delivered the best data in all cases. Concerning the TBR and TAR, the lockdown 2020 phase also had the best values (Table 4 and Figure 1). The times in different ranges in the different phases are presented for the ranges < 54 mg/dL, 54–<70 mg/dL, <70 mg/dL, 70–180 mg/dL, >180–250 mg/dL, >180 mg/dL and > 250 mg/dL. Note that some of these ranges overlap.

The differences in time in the respective glucose ranges during lockdown 2020 vs. the two other phases were evaluated as well. In all glucose ranges, pre-lockdown 2020 showed more deviations from the lockdown 2020 phase compared to the pre-pandemic 2019 phase (Table 5).

Further, the total number of daily scans and the mean number of daily scans during each phase were evaluated. We found differences to the pre-pandemic 2019 phase. The total scan frequency and the mean daily isCGM scan frequencies in this phase were higher compared to the other phases (Table 6 and Figure 2). The mean number of scans was the highest during the pre-pandemic 2019 phase, as was the total number of scans.

### 3.2. Sensitivity Analysis

A total of 80 people with T1D (Appendix A) were included in the sensitivity analysis. The diabetes duration was 12.5 years (5.5; 20.7 years), and the baseline HbA1c was at 57.5 mmol/mol (51.0; 69.3 mmol/mol). the participants had used an isCGM system for an average of 5.6 years (4.6; 6.6 years) before participating in the study and had fewer visits during lockdown than during the prior phases (Table 7).

As in the main analysis, the lockdown 2020 phase showed the best mean glucose data, with mean glucose values of 163.2 mg/dL (150.6–193.0 mg/dL). The data from 80 individuals were analyzed and in general, the isCGM parameters were similar across all three phases. The isCGM values (n) were nearly the same data in the pre-pandemic 2019 phase and in the lockdown 2020 phase and showed just a small difference during the pre-lockdown 2020 phase (Table 8).

The relative frequencies of TIR_[70–180mg/dL]_ were the highest percentages across all phases, including the sensitivity analysis, but the lockdown 2020 phase again delivered the best data in all cases. Concerning TBR and TAR, the lockdown 2020 phase also had the best values (Table 9). The percentages of relative frequencies of times in different ranges over time are presented for the ranges < 54 mg/dL, 54–<70 mg/dL, <70 mg/dL, 70–180 mg/dL, >180–250 mg/dL, >180 mg/dL and > 250 mg/dL.

Lastly, the total number of daily scans and the mean number of daily scans during each phase were also evaluated for the sensitivity analysis. We found differences compared to the pre-pandemic 2019 phase. The total scan frequency and the mean daily isCGM scan frequencies in this phase were higher compared to the other phases (Table 10). The mean number of scans was the highest during the pre-pandemic 2019 phase, as was the total number of scans. This underscores the statement of the main analysis.

## 4. Discussion

Our analysis revealed that the glycemic control assessed by isCGM was superior during the lockdown 2020 phase compared to two pre-lockdown phases. These findings might seem counter-intuitive at first because non-emergency care was reduced severely during lockdown. However, our results clearly show that the lockdown 2020 phase had the best outcomes in all parameters compared to the two other phases. We may assume that these findings are a result of having more time for individual diabetes management, for cooking healthy food and for engaging in basic endurance exercises such as running or walking instead of high-impact sports such as tennis, football or gym workouts, which were not available during lockdown. Despite the general positive aspects of physical exercise, less strenuous kinds of exercise are more beneficial in terms of diabetes outcomes [28]. Moreover, the lockdown may have led to a decrease in physical exertion in general because of more work-from-home days and fewer social activities/parties.

Our hypothesis, that lockdown provided people with more time for disease management, is supported by Schiaffini et al., who evaluated a group of 22 pre-school and school children with T1D that use basis-bolus therapy and also found better glycemic control, which was considered to be due to a stricter control of the parents regarding the glucose intake of their children [22]. Tornese et al. investigated retrospective data from 13 individuals with T1D and compared three periods (one period before the COVID-19 outbreak, one period when mobility was reduced and one period during complete lockdown). They found a higher TIR during lockdown and during mobility restrictions than during the period before the pandemic and also concluded that the restrictions due to the COVID-19 pandemic did not worsen glycemic control in T1D patients [28]. Bonora et al. observed that glucose control improved in the first week of the lockdown. They found an increase in TIR and reduced average BG in a group of 33 adult patients with T1D. Similar to the current study, they observed better results during the lockdown and argued that the increase in the available free time could have been used to prepare healthier meals and/or to follow healthier lifestyles in general [29].

Contrasting these results, Ghesquière et al. included pregnant women and compared these data (lockdown vs. pre-lockdown) in the same manner as the current study. In summary, diabetes control was worse during the COVID-19 pandemic compared to the period before [30]. Sfinari et al. also found a negative effect of the COVID-19 pandemic within their comparative study. They investigated the changes in emotional behavioral parameters of children with T1D during the COVID-19 pandemic and also took lifestyle parameters under investigation. Within their study, they found a negative influence on lifestyle parameters and the behavioral and emotional variables of those children [31].

Regarding the current analysis, the isCGM scan frequency was comparable between the lockdown 2020 phase and the pre-lockdown 2020 phase but showed a numerical increase when being compared to the pre-pandemic 2019 phase. The total scan frequencies in this phase were higher compared to the other phases. These findings could be found in both the main and sensitivity analyses. They could be explained by the fact that prior to the pandemic, patients might have been exposed to higher social pressure, with a busier lifestyle, which made them insecure about their glucose levels, thus leading to a higher scanning frequency. Given the fact that glycemic control was the best during the pandemic, when most people with T1D had to cancel their routine outpatient checks, the efficiency of professional medical care delivered in a remote manner should not be underestimated in diabetes management.

## 5. Conclusions

Our findings showed the best results during the lockdown 2020 phase. Thus, it can be stated that the effects of the COVID-19 pandemic were not entirely negative—especially for people living with T1D. However, the fact that isCGM data and glycemic control can be effectively assessed without physical visits to a medical office suggests that telemedicine offers a way to avoid physical proximity without compromising close care for patients with T1D. The COVID-19 pandemic taught the society that many systems are available remotely, saving resources and time. This includes diabetes management. It has been shown that telemedicine for diabetes could be definitely integrated in diabetes care in T1D [29] and could provide better continuity of healthcare assistance by simplifying communication [32,33,34,35]. In Austria, however, it has not (yet) been legally approved. However, telemedicine could provide better continuity of healthcare assistance by simplifying communication [36]. Regarding the impact of the COVID-19 pandemic on people in general and especially on people living with T1D, further analysis must be conducted to clarify the consequences and impacts of the pandemic.

## Figures and Tables

**Figure 1 sensors-24-04514-f001:**
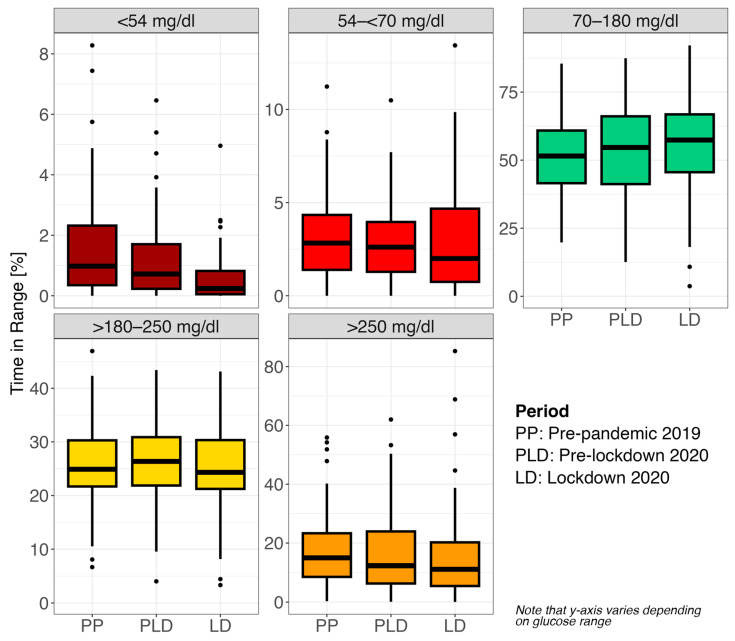
Overview of times in different (exclusive) ranges for each phase.

**Figure 2 sensors-24-04514-f002:**
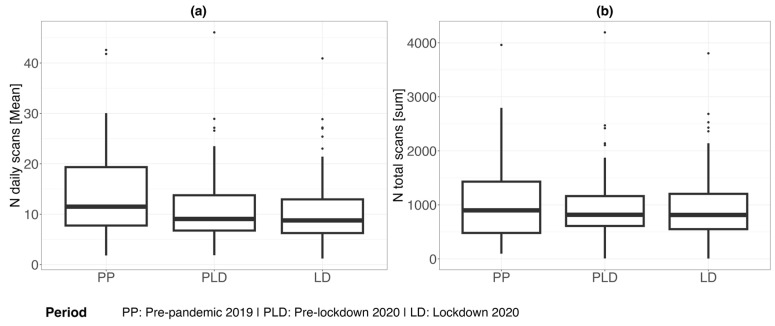
Mean (**a**) and total (**b**) number of scans during each phase.

**Table 1 sensors-24-04514-t001:** Baseline characteristics.

**Categorical parameters**	**Number (n)**	**Percent (%)**
Gender [male/female]	22 females	34
	42 males	66
System of use [pen/pump]	48 pen	75
	16 pump	25
**Numeric parameters**	**Median (Q1–Q3)**	**Min–Max**
Age [years]	33.5 (26.3; 49.5)	19.7–73.3
Diabetes duration [years]	13.5 (5.5; 22.0)	1.45–64.5
Height [m]	1.8 (1.7; 1.8)	1.6–1.9
Weight [kg]	76.0 (68.0; 88.3)	49.0–134.0
BMI [kg/m2]	24.6 (22.3; 27.0)	18.4–37.9
HbA1c [mmol/mol]	58.5 (51.0; 69.3)	37.0–100.0
Creatinine [mg/dL]	0.9 (0.8; 1.0)	0.6–4.4
isCGM use duration [months]	68.5 (58.0; 79.3)	51.0–92.0
isCGM use duration [years]	5.7 (4.8; 6.6)	4.3–7.7
Visits during lockdown 2020 [n]	0.0 (0.0; 1.0)	0.0–6.0
Visits during pre-lockdown [n]	1.0 (0.0; 1.0)	0.0–6.0
Visits during pre-pandemic [n]	1.0 (0.0; 2.3)	0.0–11.0

(isCGM—intermittently scanned continuous glucose monitoring).

**Table 2 sensors-24-04514-t002:** The isCGM-derived parameters during the three phases of observation.

Phase	Individuals (N)	Parameters	Median (Q1–Q3)	Min–Max
Pre- pandemic 2019	64	Days (n)	93.0 (85.0–93.0)	5.0–93.0
		isCGM values (n)	8311.0 (5804.5–8677.5)	399.0–9177.0
		isCGM activity (%)	95.2 (88.1–97.7)	44.8–102.8
		Mean glucose (mg/dL)	176.5 (155.9–196.1)	117.6–275.8
		CV	0.4 (0.4–0.4)	0.2–0.5
		GMI	7.5 (7.0–8.0)	6.1–9.9
		MAGE	187.3 (170.5–207.9)	123.9–286.1
Pre- lockdown 2020	64	Days (n)	91.0 (85.5–91.0)	4.0–91.0
		isCGM values (n)	8009.0 (6875.0–8357.0)	143.0–10312.0
		isCGM activity (%)	94.40 (87.2–95.9)	37.2–118.0
		Mean glucose (mg/dL)	168.4 (154.9–197.0)	113.9–299.4
		CV	0.4 (0.3–0.4)	0.2–0.5
		GMI	7.3 (7.0–8.0)	6.0–10.5
		MAGE	177.0 (165.4–205.6)	121.3–358.0
Lockdown 2020	64	Days (n)	93.0 (90.0–93.0)	6.0–93.0
		isCGM values (n)	8308.0 (7245.5–8480.0)	155.0–12934.0
		isCGM activity (%)	93.3 (85.6–95.2)	26.9–144.9
		Mean glucose (mg/dL)	163.6 (154.0–193.0)	112.4–352.0
		CV	0.4 (0.3–0.4)	0.2–0.6
		GMI	7.2 (7.0–7.9)	6.0–11.7
		MAGE	177.5 (163.5–202.4)	119.0–357.8

(CV—coefficient of variability, GMI—glucose management indicator, isCGM—intermittently scanned continuous glucose monitoring, MAGE—mean amplitude of glycemic excursion).

**Table 3 sensors-24-04514-t003:** Differences in isCGM-derived parameters between lockdown 2020 and the other phases.

Difference *	Parameter	Median (Q1–Q3)	Min–Max
Lockdown 2020 vs. pre-pandemic 2019	Days (n)	2.0 (2.0–2.0)	−56.0–66.0
	isCGM values (n)	185.0 (−2.0–577.5)	−5645.0–4181.0
	isCGM activity (%)	0.0 (−1.7–1.6)	−70.5–27.1
	Mean glucose (mg/dL)	−3.8 (−13.5–2.6)	−61.6–66.9
	CV	−0.0 (−0.0–0.0)	−0.1–0.1
	GMI	−0.1 (−0.3–0.1)	−1.5–1.6
	MAGE	−2.3 (−11.8–3.7)	−75.0–80.8
Lockdown 2020 vs. pre-lockdown 2020	Days (n)	0.0 (0.0–1.5)	−87.0–88.0
	isCGM values (n)	−42.0 (−471.0–1247.0)	−8701.0–7699.0
	isCGM activity (%)	−1.0 (−5.9–1.2)	−72.3–42.2
	Mean glucose (mg/dL)	−3.6 (−17.2–6.3)	−42.9–83.0
	CV	−0.0 (−0.1–0.0)	−0.2–0.1
	GMI	−0.1 (−0.4–0.2)	−1.0–2.0
	MAGE	−6.5 (−17.1–8.7)	−43.3–94.9

* Difference is calculated by subtracting parameters of the other phases from lockdown 2020. (CV—coefficient of variability, GMI—glucose management indicator, isCGM—intermittently scanned continuous glucose monitoring, MAGE—mean amplitude of glycemic excursion).

**Table 4 sensors-24-04514-t004:** Times in different ranges.

Phase	Glucose Range	Median (Q1–Q3)	Min–Max
Pre-pandemic 2019	<54 mg/dL	1.0 (0.4–2.3)	0.0–8.3
	54–<70 mg/dL	2.8 (1.4–4.3)	0.0–11.2
	<70 mg/dL	3.6 (1.8–6.5)	0.0–14.9
	70–180 mg/dL	51.5 (41.5–60.9)	19.8–85.5
	>180–250 mg/dL	24.9 (21.7–30.3)	6.7–47.0
	>180 mg/dL	44.6 (31.5–55.2)	7.2–80.0
	>250 mg/dL	15.0 (8.6–23.3)	0.3–55.9
Pre-lockdown 2020	<54 mg/dL	0.72 (0.2–1.7)	0.0–6.5
	54–<70 mg/dL	2.61 (1.3–4.0)	0.0–10.5
	<70 mg/dL	3.6 (1.6–5.6)	0.0–15.2
	70–180 mg/dL	54.7 (41.2–66.1)	12.6–87.4
	>180–250 mg/dL	26.4 (21.9–30.9)	4.0–43.4
	>180 mg/dL	40.6 (30.0–56.1)	4.5–87.4
	>250 mg/dL	12.3 (6.3–24.0)	0.1–62.0
Lockdown 2020	<54 mg/dL	0.2 (0.1–0.8)	0.0–5.0
	54–<70 mg/dL	2.0 (0.7–4.7)	0.0–13.4
	<70 mg/dL	2.3 (0.8–6.1)	0.0–14.9
	70–180 mg/dL	57.4 (45.6–66.8)	3.7–92.2
	>180–250 mg/dL	24.3 (21.2–30.3)	3.4–43.1
	>180 mg/dL	35.4 (28.9–53.3)	3.4–96.3
	>250 mg/dL	11.2 (5.5–20.2)	0.1–85.3

**Table 5 sensors-24-04514-t005:** Differences in time in respective ranges.

Difference *	Parameter	Median (Q1–Q3)	Min–Max
Lockdown 2020 vs. pre-pandemic 2019	<54 mg/dL	−0.4 (−1.0–−0.1)	−3.7–1.4
	54–<70 mg/dL	−0.1 (−0.6–1.1)	−4.9–4.9
	<70 mg/dL	−0.3 (−1.4–0.7)	−6.3–6.3
	70–180 mg/dL	3.9 (−1.6–7.1)	−19.0–23.7
	>180–250 mg/dL	−1.2 (−3.8–1.0)	−16.8–18.1
	>180 mg/dL	−3.0 (−7.3–1.0)	−30.0–20.0
	>250 mg/dL	−0.9 (−4.1–0.7)	−26.4–23.3
Lockdown 2020 vs. pre-lockdown 2020	<54 mg/dL	−0.6 (−1.5–−0.1)	−5.0–1.9
	54–<70 mg/dL	−0.2 (−1.1–0.6)	−6.4–6.4
	<70 mg/dL	−1.2 (−2.4–0.1)	−7.4–5.9
	70–180 mg/dL	3.2 (−1.7–9.7)	−20.0–27.7
	>180–250 mg/dL	−0.5 (−3.2–1.9)	−16.7–19.3
	>180 mg/dL	−2.0 (−10.0–2.1)	−27.7–21.0
	>250 mg/dL	−1.8 (−5.2–0.9)	−17.0–29.4

* Difference is calculated by subtracting parameters of the other phases from lockdown 2020.

**Table 6 sensors-24-04514-t006:** isCGM scan frequencies.

Parameter	Phase	Median (Q1–Q3)	Min–Max
Total isCGM scan frequency	Pre-pandemic 2019	899.0 (481.0–1430.5)	97.0–3961.0
	Pre-lockdown 2020	817.0 (610.0–1163.0)	9.0–4193.0
	Lockdown 2020	814.0 (551.0–1204.5)	7.0–3805.0
Mean daily isCGM scan frequency	Pre-pandemic 2019	11.5 (7.8–19.4)	1.8–42.6
	Pre-lockdown 2020	9.1 (6.8–13.8)	1.9–46.1
	Lockdown 2020	8.8 (6.3–13.0)	1.2–41.0

(isCGM—intermittently scanned continuous glucose monitoring).

**Table 7 sensors-24-04514-t007:** Sensitivity analysis: baseline characteristics.

**Categorical parameters**	** Number (n) **	** Percent (%) **
Gender [male/female]	26 females	32.5
	54 males	67.5
System of use [pen/pump]	60 pen	75
	20 pump	25
** Numeric parameters **	** Median (Q1–Q3) **	** Min–Max **
Age [years]	33.2 (25.5; 49.1)	19.2–73.3
Diabetes duration [years]	12.5 (5.5; 20.7)	1.5–64.5
Height [m]	1.8 (1.7; 1.8)	1.5–1.9
Weight [kg]	76.0 (68.0; 88.3)	43.0–134.0
BMI [kg/m^2^]	24.0 (22.0; 26.9)	17.4–37.9
HbA1c [mmol/mol]	57.5 (51.0; 69.3)	37.0–143.0
Creatinine [mg/dL]	0.9 (0.8; 1.0)	0.6–4.4
isCGM use duration [months]	67.0 (55.0; 79.0)	47.0–92.0
isCGM use duration [years]	5.6 (4.6; 6.6)	3.9–7.7
Visits during lockdown 2020 [n]	0.0 (0.0; 1.0)	0.0–6.0
Visits during pre-lockdown [n]	1.0 (0.0; 1.0)	0.0–7.0
Visits during pre-pandemic) [n]	1.0 (0.0; 2.0)	0.0–11.0

(isCGM—intermittently scanned continuous glucose monitoring).

**Table 8 sensors-24-04514-t008:** Sensitivity analysis: isCGM-derived parameters.

Phase	Individuals (N)	Parameters	Median (Q1–Q3)	Min–Max
Pre- pandemic 2019	64	Days (n)	93.0 (85.0–93.0)	5.0–93.0
		isCGM values (n)	8309.0 (5829.8–8675.3)	399.0–9177.0
		isCGM activity (%)	95.2 (87.9–97.6)	44.8–102.8
		Mean glucose (mg/dL)	177.7 (156.2–195.7)	117.6–275.8
		CV	0.4 (0.4–0.4)	0.2–0.5
		GMI	7.6 (7.1–8.0)	6.1–9.9
		MAGE	187.5 (170.8–207.8)	123.9–286.1
Pre- lockdown 2020	78	Days (n)	91.0 (82.5–91.0)	4.0–91.0
		isCGM values (n)	7930.0 (6646.3–8344.5)	143.0–10312.0
		isCGM activity (%)	93.32 (83.7–95.8)	37.2–118.0
		Mean glucose (mg/dL)	166.7 (154.0–195.3)	107.8–299.4
		CV	0.4 (0.3–0.4)	0.2–0.5
		GMI	7.3 (7.0–8.0)	5.9–10.5
		MAGE	175.4 (164.3–203.9)	121.3–358.0
Lockdown 2020	79	Days (n)	93.0 (90.0–93.0)	1.0–93.0
		isCGM values (n)	8283.0 (7349.5–8478.0)	32.0–12934.0
		isCGM activity (%)	92.83 (85.6–95.0)	26.9–144.9
		Mean glucose (mg/dL)	163.2 (150.6–193.0)	110.4–352.0
		CV	0.4 (0.3–0.4)	0.2–0.6
		GMI	7.2 (6.9–7.9)	6.0–11.7
		MAGE	175.7 (162.4–197.6)	119.0–357.8

(CV—coefficient of variability, GMI—glucose management indicator, isCGM—intermittently scanned continuous glucose monitoring, MAGE—mean amplitude of glycemic excursion).

**Table 9 sensors-24-04514-t009:** Sensitivity analysis: time in different ranges.

Phase	Glucose Range	Median (Q1–Q3)	Min–Max
Pre-pandemic 2019	<54 mg/dL	0.9 (0.4–2.3)	0.0–8.3
	54–<70 mg/dL	2.8 (1.5–4.3)	0.0–11.2
	<70 mg/dL	3.6 (1.9–6.5)	0.0–14.9
	70–180 mg/dL	51.6 (41.7–60.8)	19.8–85.5
	>180–250 mg/dL	25.2 (21.7–30.3)	6.7–47.0
	>180 mg/dL	44.6 (31.5–55.1)	7.2–80.0
	>250 mg/dL	15.3 (8.7–23.2)	0.3–55.9
Pre-lockdown 2020	<54 mg/dL	0.7 (0.3–1.8)	0.0–7.7
	54–<70 mg/dL	2.8 (1.4–4.3)	0.0–13.9
	<70 mg/dL	3.7 (1.7–6.1)	0.0–18.7
	70–180 mg/dL	55.7 (41.9–67.5)	12.6–90.6
	>180–250 mg/dL	24.7 (21.0–29.7)	4.0–43.4
	>180 mg/dL	38.7 (29.3–55.8)	4.5–87.4
	>250 mg/dL	11.6 (4.6–24.1)	0.1–62.0
Lockdown 2020	<54 mg/dL	0.2 (0.1–0.8)	0.0–5.0
	54–<70 mg/dL	2.0 (0.8–5.4)	0.0–13.4
	<70 mg/dL	2.3 (0.8–6.4)	0.0–14.9
	70–180 mg/dL	58.4 (45.4–68.4)	3.7–92.2
	>180–250 mg/dL	24.3 (20.7–28.5)	3.4–43.1
	>180 mg/dL	35.3 (27.9–52.9)	3.4–96.3
	>250 mg/dL	10.0 (4.9–20.7)	0.1–85.3

**Table 10 sensors-24-04514-t010:** Sensitivity analysis: isCGM scan frequencies.

Parameter	Phase	Median (Q1–Q3)	Min–Max
Total isCGM scan frequency	Pre-pandemic 2019	900.0 (483.5–1426.8)	97.0–3961.0
	Pre-lockdown 2020	810.5 (553.8–1172.0)	9.0–4193.0
	Lockdown 2020	816.0 (567.0–1261.0)	2.0–3805.0
Mean daily isCGM scan frequency	Pre-pandemic 2019	11.6 (7.8–19.3)	1.8–42.6
	Pre-lockdown 2020	9.2 (6.7–14.7)	1.9–46.1
	Lockdown 2020	8.9 (6.4–13.6)	1.2–40.9

(isCGM—intermittently scanned continuous glucose monitoring).

## Data Availability

Data can be requested upon reasonable analysis of the idea.

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
