# Peer review of "Glycemic Control Assessed by Intermittently Scanned Glucose Monitoring in Type 1 Diabetes during the COVID-19 Pandemic in Austria"

_sensors, 2024, doi:10.3390/s24144514_

Round 1
Reviewer 1 Report
Comments and Suggestions for Authors
The manuscript “Glycemic control assessed by intermittently scanned glucose monitoring in type 1 diabetes during the COVID-19 pandemic” describes changes or lack of changes in the characteristics of type 1 diabetes patients during the COVID-19 pandemic. Authors investigated levels of intermittently scanned continuous glucose monitoring (isCGM), isCGM activity, hemoglobin A1c (HbA1c), mean glucose, mean amplitude of glycemic excursion (MAGE), glucose management indicator (GMI), coefficient of variability (CV) in pre-pandemic, pre-lockdown, and lockdown 2020. The data presented in the article are purely statistical.
The manuscript is relevant for the field but should be presented in a better structured manner. There are should be more explanations in the Results section. There are only tables with average values of characteristics of diabetes type 1 in different time period of COVID-19. What scientific meaning does these results have? There is very short Discussions section and absence Conclusions section.
A more extensive Discussion and a highlighted Conclusion chapters would help to give the publication scientific significance. In this version, the manuscript does not correspond to the high rating of the journal.
Reviewer 2 Report
Comments and Suggestions for Authors
The manuscript “Glycemic control assessed by intermittently scanned glucose monitoring in type 1 diabetes during the COVID-19 pandemic” describes an intermittently scanned continuous glucose monitoring to evaluate the glucose levels in people at pre-pandemic, pre-lockdown and lockdown (2020) in Austria. Below are some suggestions for improving the work before publication:
I a) I think the novelty of the article should be better highlighted by the authors.
2b) I believe that the article's relationship with the scope of Sensors journal should be better highlighted. It is not possible to understand the relationship between the theme and sensors or Internet of Things.
3c) In Page 1, line 42, the authors wrote “development of micro-and macrovascular complications and diseases”. Which diseases? I think it is important to include related diseases.
4d) There are several acronyms in the manuscript that have no definition, or that the definition is long after the acronym is mentioned. I suggest a good review.
5e) The introduction section is concise. I think more information is important to improve understanding of the novelty of the research, as the use of intermittently scanned continuous glucose monitoring (isCGM) and parameters analysed.
6f) Table 2 has a caption “. isCGM parameters” but other parameters is presented.
7g) Table 4 present “time in different ranges” but it is not possible to understand which time. The same to Table 5.
8h) Is the analyzed data exclusively from Austria? If so, I think it would be interesting to make it clear in the title of the manuscript
Round 2
Reviewer 1 Report
Comments and Suggestions for Authors
The authors made significant changes to the manuscript “Glycemic control assessed by intermittently scanned glucose monitoring in type 1 diabetes during the COVID-19 pandemic”. Sections of the Materials and Methods, Results and Discussion are expanded. There is Conclusion section now, that make the results a little bit clear.
The current version of the manuscript is clear, relevant for the field and presented in a well-structured manner. The cited references include recent publications (within the last 5 years) and are relevant. It does not include an excessive number of self-citations. The manuscript scientifically sound and the experimental design is appropriate to test the hypothesis. The figures and tables are appropriate and properly show the data.
There is just one note: inscriptions on the figures (axis names etc.) are not readable, it should be increased in size.
Author Response
Comment 1: There is just one note: inscriptions on the figures (axis names etc.) are not readable, it should be increased in size.
Response 1: Thank you again for your feedback and for making us alert, that the inscriptions on the figures are not readable. We increased them in size and made the inscriptions more clear. Please find the latest version attached.
Reviewer 2 Report
Comments and Suggestions for Authors
The suggestions were inserted in the manuscript.
Author Response
Thank you very much for your positive feedback and your assumption, that all suggestions were inserted in the script.
We conducted some last small minor changes. Please find the latest version attached.